# Prevalence of anxiety and depression during COVID-19 pandemic among healthcare students in Jordan and its effect on their learning process: A national survey

**Iman A. Basheti**[1]*, **Qassim N. Mhaidat**[2], **Hala N. Mhaidat**[2]

**1** Department of Clinical Pharmacy and Therapeutics, Faculty of Pharmacy, Applied Science Private University, Amman, Jordan, **2** School of Medicine, Jordan University of Science and Technology, Irbid, Jordan

* dr_iman@asu.edu.jo

## Abstract

### Rational

During pandemics, including the most recent COVID-19 pandemic, the mental health of university healthcare students' is expected to be affected negatively, impacting the students' learning process.

### Objectives

The aim of this study was to assess the level of anxiety and depression of healthcare students living in Jordan, and the effect on their learning process during the COVID-19 pandemic.

### Methods

This descriptive cross-sectional study was conducted via an online survey completed by students studying a healthcare-oriented degree in a university in Jordan. Participants were recruited through social media (Facebook and WhatsApp). The validated previously published Hospital Anxiety and Depression Scale (HADS) questionnaire was used as a part of the online survey to assess students' anxiety/depression scores. Students' responses regarding their learning process during the COVID-19 was also assessed.

### Results

The mean age of participants was 21.62 (SD = 4.90), with the majority being females (67.1%). The HADs' assessment revealed that 43.8% and 40.0% of participants had normal anxiety and depression scores, while 22.4% showed borderline abnormal anxiety/depression scores (33.8%). Many students (33.8%) were classified to have abnormal anxiety scores, while a smaller proportion (26.2%) was classified to have abnormal depression scores. Smoking (p = 0.022), lower family income (p = 0.039), and use of medications (p = 0.032) were positively associated with higher (worse) anxiety scores. Ranking the learning

**Data Availability Statement:** All relevant data are within the paper and its Supporting information files.

**Funding:** The authors received no specific funding for this work.

**Competing interests:** The authors have declared that no competing interests exist.

process during COVID-19 showed that 45.8% of the participants believed it was a 'good/very good/excellent' process.

## Conclusions

Anxiety and depression levels amongst university healthcare students in Jordan were found to be high when assessed during the COVID-19 pandemic. In addition, the learning process during the pandemic was not accepted by more than half of the students. Implementing psychological interventions for healthcare students during pandemics is strongly recommended in order to optimize students' mental health and their learning process alike.

## Introduction

Since December 2019, the outbreak of the coronavirus disease has been spreading rapidly from China to other parts of the world leading to acute infectious pneumonia [1]. In March 2020, the World Health Organization (WHO) announced the corona virus disease, termed COVID-19 disease, as an international pandemic, which was caused by the infectious virus 'severe acute respiratory syndrome (SARS) coronavirus 2' [2, 3]. Similar to the previous 2002 and 2012 viral outbreaks, SARS and Middle East Respiratory Syndrome (MERS-CoV), COVID-19 symptoms included breathing difficulties accompanied with fever and coughing [4, 5]. Although severe in the symptoms displayed, no established antiviral treatment or vaccine has been approved for the management of COVID-19 to this date [6, 7]. On July 29, 2020, the Jordan Ministry Of Health reported that the COVID-19 outbreak has resulted in 1,187 confirmed cases in Jordan, and 11 cases of confirmed deaths [8].

Social and physical distancing, in addition to self-quarantining, have been imposed by many governments worldwide, due to the spread of the virus reported to be primarily through direct contact, i.e. droplets spread by coughing or sneezing from an infected individual [9]. Quarantine is an effective measure in decreasing the spread of the virus globally [10], however, it comes with numerous substantial economic, social, and psychological effects [11]. In consequence to this, several challenges and concerns, including psychological pressures, have been enacted on individuals [6, 12].

Mental health of individuals is a major health concern, expected to be disturbed during pandemics, including the COVID-19 pandemic [13]. According to previous research reports, during similar viral outbreaks, a significant increase in the risk of mental health problems among individuals happen, including anxiety, depression and traumatic stress [14–17]. During the recent COVID-19 pandemic, increased levels of stress, anger, anxiety and depression have been reported among individuals in different parts of the world [11, 18, 19].

In an effort to halt the spread of the virus, most governments took several safety measures varying from temporary postpone of activities and events in educational institutions to a complete closure of schools and universities [20, 21]. Distance learning became the route of education implemented in most countries, and unsurprisingly, enclosing new concerns and challenges for students [21–23]. Thus, as a consequence to this major change from the norm, mental health of college students was expected to be affected [21].

Anxiety and depression have been reported to be caused by public emergencies including pandemics, affecting the mental health of people, including college students [24, 25]. It has been acknowledged that COVID-19 pandemic significantly affected the educational process, career progression, health and safety of medical students [26]. More concerns have been

brought to light regarding medical students' education as a consequence to the impact of COVID-19. No previous study has looked into the precise effect of the COVID-19 pandemic on the mental health of medical students in Jordan, nor abroad, and its effect on their learning process.

The aim of this study was to assess the level of anxiety and depression of medical students living in Jordan, and its effect on their learning process during the COVID-19 pandemic.

## Methods

### Study design and participants

The study objectives were addressed using a descriptive cross-sectional online survey distributed to healthcare students studying in Jordan via the social media (Facebook and WhatsApp). This study was conducted during the Coronavirus outbreak (July 14th to July 29th, 2020). Following development by the research team, the online survey was tested during a pilot phase of this study, which was conducted over three days before the actual study data collection period was initiated. The online survey was completed by students (n = 5) and academics (n = 5) who gave their feedback and recommendations regarding the survey. The research team studies these recommendations and feedback provided, and a final version of the survey was prepared before the study started. Results of the pilot phase of the study were not included in the analysis of this study.

Eligible participants were students studying a healthcare-oriented degree (medicine, dentistry, Pharm.D., pharmacy, nursing, and other) in a university, including both public and private universities. Ethics approval for the study was obtained from the Faculty of Pharmacy, Applied Science Private University (2020-PHA-19). It was made clear to the participants that study participation did not pose any risk to them and was voluntary. Potential participants who completed the survey were considered to have given informed consent for study participation.

### Study tool

The first part of the online survey included information regarding the participants' demographic data, university type, nationality, level of study, social status, socio-economic status, smoking status, caffeine intake, and medication/s use. The second section of the first part verified if a family member was previously diagnosed with anxiety and/or depression, participants' sleeping pattern hours during COVID-19, actual sleeping hours during COVID-19, and changes in participants' sleep pattern during COVID-19.

The second part of the survey was set to evaluate participants' anxiety and depression status, using the validated and previously published Hospital Anxiety and Depression Scale (HADS) [27]. In assessing the anxiety status, the following questions were included in the survey: "*I feel tense or wound up; I get a sort of frightened feeling as if something awful is about to happen; Worrying thoughts go through my mind; I can sit at ease and feel relaxed; I get a sort of frightened feeling like 'butterflies' in the stomach; I feel restless as I have to be on the move and I get sudden feelings of panic*".

As for the assessment of depression status, the following questions were asked: "*I still enjoy the things I used to enjoy; I can laugh and see the funny side of things; I feel cheerful; I feel as if I am slowed down; I have lost interest in my appearance; I look forward with enjoyment to things; I can enjoy a good book or radio or TV program*".

Each item in HADS survey was rated on a four-point scale, giving maximum score of 21. Scores were divided into three categories: normal (0–7), borderline abnormal (8–10), and abnormal (11–21) cases of anxiety and depression [28].

The third part of the online survey assessed students believes regarding their learning process during the COVID-19 pandemic. The following items were included in this part of the survey: "*how do you rank the online learning with regards to theory courses you received last semester compared to last year; practical courses you received last semester compared to last year; assessments and exams you received last semester compared to last year training last semester compared to last year; your relationship with your doctors last semester compared to last year's; and effect on your last semester (second semester of 2020) Grade Point Average (GPA) compared to last year's second semester (second semester of 2019)*".

## Survey implementation

Study participants who chose to participate opened a link to initially view ethics committee approved information about the study, and then proceeding with completing the survey via the Facebook or the WhatsApp social media applications.

## Sample size

According to the Ministry of Higher Education and Scientific Research, the number of the healthcare students in Jordan in 2020 was 16, 214 [29]. Based on this number, the sample size was calculated using a margin of error of 5%, confidence level of 95%, and response distribution of 50%, to be a minimum of 375 students.

## Statistical analysis

Data were analysed using Statistical Package for Social Science (SPSS) version 24 (SPSS Inc., Chicago, IL, USA). Checking for data normality was carried out using the Shapiro-Wilk test (with P-value $\geq 0.05$ indicating a normally distributed continuous variable). Study primary outcomes including anxiety and depression mean scores, and anxiety and depression severity levels, were assessed using descriptive statistics and frequencies. All continuous variables were expressed as mean (SD), and analysed using the Independent Sample t-test and Paired Sample t-Test for comparison purposes, while categorical variables were expressed as proportions (%) and analysed using Pearson's Chi-Square test and McNemar-Bowker test. All p values of less than 0.05 were considered statistically significant.

In order to determine predictors of the dependent variables, anxiety and depression scores, a multiple linear regression analysis was performed after an assessment for collinearity. Associations between the dependent variable, anxiety mean score (Model A) and depression mean score (Model B) were analysed against the independent variables: smoking status during COVID-19 (none, social smoker (less than 10 cigs a day, and/or not more than 3 shisha a week), smoker (more than 10 cigs a day, and/or more than 3 shisha a week)), living place (city, rural), nationality (Jordanian, Non-Jordanian), age, gender (female, male), family monthly income level ($< 500$ Jordanian Dinar (JD), 500–1000 JD, $> 1000$ JD, $> 2000$ JD, $> 3000$ JD), caffeine intake during COVID-19 (none, less than 2 cups, 2–4 cups, more than 4 cups), and use of medications (antihistamine, non-steroidal anti-inflammatory drugs (NSAIDs), paracetamol, multivitamins, muscle relaxants, puffers, others).

## Result

### Demographic characteristics

Study participants (n = 450) were from public (mainly Jordan University of Science and Technology (42.4%) and Yarmouk University (24.4%)) and private universities (mainly Applied Science Private University (22.6%)) across Jordan. Demographic data (Table 1) showed a

**Table 1. Demographic characteristics of the study sample (n = 450).**

| Parameter | n (%) |
|---|---|
| Age, (mean ±SD) | 21.62 ±4.9 |
| Gender, n (%) | |
| • Female | 302 (67.1) |
| • Male | 148 (32.9) |
| University, n (%) | |
| • Jordan University of Science and Technology | 188 (42.4) |
| • Applied Science Private University | 100 (22.6) |
| • Yarmouk University | 108 (24.4) |
| • The University of Jordan | 7 (1.6) |
| • The Hashemite University | 8 (1.8) |
| • Others | 32 (7.2) |
| Study Field, n (%) | |
| • Medicine | 201 (44.7) |
| • Dentistry | 34 (7.6) |
| • Pharm.D | 10 (2.2) |
| • Pharmacy | 146 (32.4) |
| • Nursing | 43 (9.6) |
| • Others | 16 (3.6) |
| Level of study, n (%) | |
| • Higher education–PhD | 12 (2.7) |
| • Higher education–Masters | 28 (6.2) |
| • Undergraduate education | 410 (91.1) |
| Nationality, n (%) | |
| • Jordanian | 352 (78.2) |
| • Non-Jordanian | 98 (21.8) |
| Jordanian, n (%) | |
| • Amman | 131 (35.9) |
| • Irbid | 174 (47.7) |
| • Others | 60 (16.4) |
| Social Status, n (%) | |
| • Single | 423 (94) |
| • Married | 26 (5.8) |
| • Divorced | 1 (0.2) |
| Children, n (%) | |
| • Yes | 23 (5.1) |
| • No | 427 (94.9) |
| Family Monthly Income, n (%) | |
| • Less than 500 JD | 72 (16) |
| • 500–1000 JD | 155 (34.4) |
| • More than 1000 JD | 90 (20) |
| • More than 2000 JD | 76 (16.9) |
| • More than 3000 JD | 57 (12.7) |
| Living place, n (%) | |
| • City | 360 (80) |
| • Rural | 90 (20) |
| Smoking during COVID-19, n (%) | |

*(Continued)*

**Table 1.** (Continued)

| Parameter | n (%) |
|---|---|
| • None | 366 (81.3) |
| • Social smoker (less than 10 cigs a day, and/or not more than 3 shisha a week) | 51 (11.3) |
| • Smoker (more than 10 cigs a day, and/or more than 3 shisha a week) | 33 (7.3) |
| Caffeine intake during COVID-19, n (%) | |
| • None | 130 (28.9) |
| • Less than 2 cups | 219 (48.7) |
| • 2–4 cups | 84 (18.7) |
| • More than 4 cups | 17 (3.8) |
| Use of medication/s, n (%) | |
| • None | 290 (65.6) |
| • Antihistamine | 28 (6.3) |
| • NSAIDs | 8 (1.8) |
| • Paracetamol | 35 (7.9) |
| • Multivitamins | 40 (9.0) |
| • Muscle relaxants | 12 (2.7) |
| • Puffers | 1 (0.2) |
| • Others (thyroxin, carbamazepine, sumatriptan, tramaldol, roaccutan, PPI, probiotics, metformin, insulin) | 28 (6.3) |
| Family member diagnosed with anxiety and/or depression, n (%) | |
| • No | 357 (80.8) |
| • Yes, from my mother's side | 28 (6.3) |
| • Yes, from my father's side | 35 (7.9) |
| • Yes, from both my parent's side | 22 (5.0) |
| Changes in sleep pattern during COVID-19, n (%) | |
| • Yes | 353 (78.4) |
| • No | 97 (21.6) |
| Sleeping hours during COVID-19, n (%) | |
| • Less than 4 hours | 17 (3.8) |
| • 4–6 hours | 59 (13.1) |
| • 6–8 hours | 177 (39.3) |
| • More than 8 hours | 197 (43.8) |
| Sleeping pattern during COVID-19, n (%) | |
| • I sleep during the day only | 63 (14) |
| • I sleep during the night only | 146 (32.4) |
| • I sleep during the day and night equally | 74 (16.4) |
| • I sleep during the day and nap during the night | 53 (11.8) |
| • I sleep during the night and nap during the day | 114 (25.3) |

COVID-19 = coronavirus disease 2019; NSAIDs = None-steroidal anti-inflammatory drugs; PPI = proton pump inhibitors.

mean age of 21.62 (SD = 4.9), with most of the students being females (67.1%). Majority of the participants were undergraduate (91.1%), Jordanians (78.2%), medicine students (44.7%), single (94%), non-smokers (81.3%), living in a city type zone (80.0%) and reporting a family income between 500 and 1000 Jordanian Dinar (JD; 34.4%). Most participants were not taking any medications (65.6%), while others were taking over the counter medications (OTCs) including multivitamins, paracetamol, antihistamines, muscle relaxants, NSAIDs and puffers.

Only few (6.3%) of the participants reported were taking prescribed medications, including thyroxin, carbamazepine, sumatriptan, tramadol, isotretinoin, proton pump inhibitors (PPI), probiotics, metformin, and insulin.

Results concerning COVID-19 pandemic period and participants' lifestyle, most participants reported to drink less than 2 cups of coffee a day (48.7%). Many (78.4%) reported a change in their sleeping pattern, with around 40.0% reported to sleep 6 to 8 hours a day and only 32.4% were just sleeping during the night.

Most of the participants reported that none of their family members were diagnosed with anxiety and/or depression (80.8%); only 5.0% of them reported to have a family member from both their parental sides previously diagnosed with anxiety and/or depression.

## Assessment of participants' anxiety

Assessing participants' anxiety (Table 2) showed that 39.8% of the participants reported feeling tensed or wounded up from time to time (occasionally), while 22.7% experienced that a lot of the time. Many (34.7%) felt frightened as if something awful was about to happen, but 'not too badly'; while 33.3% felt frightened 'a little' as if something awful was about to happen but it did not worry them. Many (37.8%) had worrying thoughts go through their mind from time to time, while others (23.8%) experienced it a lot of the time.

As for sitting at ease and feeling relaxed, only 40.0% responded with 'usually'; others declared that 'not often' did they sit at ease and feel relaxed (32.9%). More than half of the students (54.7%) reported not feeling frightened or to have a feeling like 'butterflies' in their stomach, while many others (23.6%) experienced that fearful feeling 'occasionally'.

Only 14.7% of the participants reported feeling restless and felt on the move. Regarding having a sudden feeling of panic attacks, few (9.3%) reported to have experienced that 'often' or 'quite often' (19.8%).

## Assessment of participants' depression

When participants were asked about enjoying the things they used to enjoy previously, only 36.2% reported not to feel the same enjoyment 'quite so much'. Only half of the students (50.4%) reported to be able to laugh and see the funny side of things 'as much as they always could'. A different insight into participants' feelings revealed that only 46.9% of them 'sometimes' felt cheerful, while others 'not often/not at all' did they feel cheerful (24.2%, 13.8% respectively).

With regards to feeling slowed down, 42.2% reported 'sometimes' feeling slowed down, while others reported to 'very often/nearly all the time' felt slowed down (26.9%, 17.6% respectively).

Although young in age, 20.4% of the students lost interest in their appearance during the COVID-10 pandemic period. Others reported 'I may not take quite as much care as I should (27.3%) and 'I may not take quite as much care (27.8%).

With regards to looking forward with enjoyment to things, only 11.1% reported they 'hardly did that at all' during the pandemic; 20.0% answered with doing that 'definitely less than they used to' (35.8%). Enjoying a good book/radio/TV program by participants' showed that 37.6% of them only 'sometimes experienced enjoyment during the pandemic', while 11.1% 'felt that very seldom'.

Looking at these results from another angle, the HADS online survey scales' assessment showed that two third of the participants had normal anxiety and depression scores, while 22.4% of them showed borderline abnormal anxiety scores, while 33.8% showed borderline

**Table 2. Students' anxiety and depression assessment (n = 450) via the Hospital Anxiety and Depression Scale (HADS).**

| Statement | Answer, n (%) |
| --- | --- |
| **Anxiety assessment** | |
| **I feel tense or wound up** | |
| Most of the time | 92 (20.4) |
| A lot of the time | 102 (22.7) |
| From time to time, occasionally | 179 (39.8) |
| Not at all | 77 (17.1) |
| **I get a sort of frightened feeling as if something awful is about to happen** | |
| Very definitely and quite badly | 59 (13.1) |
| Yes, but not too badly | 156 (34.7) |
| A little, but it doesn't worry me | 150 (33.3) |
| Not at all | 85 (18.9) |
| **Worrying thoughts go through my mind** | |
| A great deal of the time | 68 (15.1) |
| A lot of the time | 107 (23.8) |
| From time to time, but not too often | 170 (37.8) |
| Only occasionally | 105 (23.3) |
| **I can sit at ease and feel relaxed** | |
| Definitely | 86 (19.1) |
| Usually | 180 (40.0) |
| Not Often | 148 (32.9) |
| Not at all | 36 (8.0) |
| **I get a sort of frightened feeling like 'butterflies' in the stomach** | |
| Not at all | 246 (54.7) |
| Occasionally | 106 (23.6) |
| Quite Often | 74 (16.4) |
| Very Often | 24 (5.3) |
| **I feel restless as I have to be on the move** | |
| Very much indeed | 66 (14.7) |
| Quite a lot | 107 (23.8) |
| Not very much | 178 (39.6) |
| Not at all | 99 (22.0) |
| **I get sudden feelings of panic** | |
| Very often indeed | 42 (9.3) |
| Quite often | 89 (19.8) |
| Not very often | 134 (29.8) |
| Not at all | 185 (41.1) |
| **Depression assessment** | |
| **I still enjoy the things I used to enjoy** | |
| Definitely as much | 136 (30.2) |
| Not quite so much | 165 (36.7) |
| Only a little | 103 (22.9) |
| Hardly at all | 46 (10.2) |
| **I can laugh and see the funny side of things** | |
| As much as I always could | 227 (50.4) |
| Not quite so much now | 121 (26.9) |
| Definitely not so much now | 72 (16.0) |

(*Continued*)

**Table 2.** (Continued)

| Statement | Answer, n (%) |
|---|---|
| Not at all | 30 (6.7) |
| **I feel cheerful** | |
| Not at all | 62 (13.8) |
| Not often | 109 (24.2) |
| Sometimes | 211 (46.9) |
| Most of the time | 68 (15.1) |
| **I feel as if I am slowed down** | |
| Nearly all the time | 79 (17.6) |
| Very often | 121 (26.9) |
| Sometimes | 190 (42.2) |
| Not at all | 60 (13.3) |
| **I have lost interest in my appearance** | |
| Definitely | 92 (20.4) |
| I don't take as much care as I should | 123 (27.3) |
| I may not take quite as much care | 125 (27.8) |
| I take just as much care as ever | 110 (24.4) |
| **I look forward with enjoyment to things** | |
| As much as I ever did | 161 (35.8) |
| Rather less than I used to | 149 (33.1) |
| Definitely less than I used to | 90 (20.0) |
| Hardly at all | 50 (11.1) |
| **I can enjoy a good book or radio or TV program** | |
| Often | 150 (33.3) |
| Sometimes | 169 (37.6) |
| Not often | 81 (18.0) |
| Very seldom | 50 (11.1) |

abnormal depression scores. Many participants (33.8%) were classified to have abnormal anxiety, while a smaller proportion (26.2%) was classified to have abnormal depression (Fig 1).

## Associations with anxiety and depression scores

To unveil predictors of the dependent variable anxiety score, multiple linear regression analysis showed that participants' smoking status (being a smoker; $p = 0.022$), having a lower family monthly income ($p = 0.039$), and using medications ($p = 0.032$) were positively associated with higher anxiety scores (worst anxiety levels). Use of medications ($p = 0.031$) was the only variable that showed positive association with the depression scores, being the dependent variable (Table 3).

## Assessment of students' learning process during COVID-19

The responses of the students regarding their learning process during the COVID-19 pandemic are represented in Table 4. With regards to the theory courses, few participants ranked the online learning they received in the second semester of 2019/2020 academic year, i.e. during the pandemic, as compared to the second semester of 2018/2019 to be excellent/very good (27.4%), or good (32.9%), while many thought it was fair or not good (39.8%).

With regards to the practical courses, the proportion of students who ranked the online learning as not good was higher as compared to the theory courses (46.7%). As for

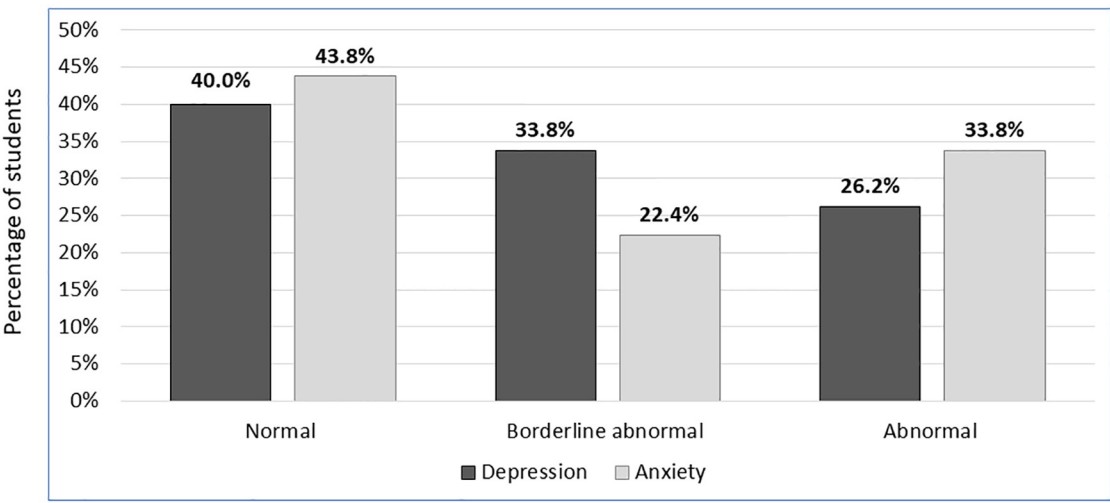

**Fig 1. Percentages of students with normal, borderline abnormal or abnormal anxiety and depression amongst study participants (n = 450).**

'assessments and exams', more than half of the participants ranked their experience as 'good, very good or excellent' (57.8%).

Ranking the learning process in general showed that 45.8% of the participants thought it was 'good or very good or excellent'. The impact of the recent learning process on the participants' relationship with their doctors showed that 52.4% of them believed it was 'good or very good or excellent'.

Regarding the semester GPA during the COVID-19 pandemic as compared to the previous year's (second semester) showed that the majority (68.0%) believed it had a positive effect. The total mark mean semester GPA was slightly increased in 2020–2021 compared to the same period of time in the 2019–2020 academic year (84.63 (SD = 11.78) vs. 83.47 (SD = 12.10)).

**Table 3. Summary of the regression model obtained for the dependent variable, students' anxiety and depression mean scores (n = 450).**

| Variable | Anxiety dependent variable | | | Depression dependent variable | | |
|---|---|---|---|---|---|---|
| | Model A | | | Model B | | |
| | Beta | t | p value | Beta | t | p value |
| Smoking during COVID_19 | .117 | 2.296 | **.022** | .096 | 1.897 | .059 |
| Living place | -.041- | -.847- | .397 | -.029- | -.602- | .547 |
| Nationality | .050 | 1.032 | .303 | .055 | 1.140 | .255 |
| Age | -.075- | -1.550- | .122 | .027 | .550 | .583 |
| Gender | .029 | .584 | .560 | -.022- | -.434- | .664 |
| Family monthly income level | -.099- | -2.067- | **.039** | -.134- | -2.795- | .005 |
| Caffeine intake during COVID_19 | .063 | 1.318 | .188 | .070 | 1.475 | .141 |
| Use of medications | .101 | 2.147 | **.032** | .102 | 2.164 | **.031** |

This table shows the output from a multivariable regression analysis in which Anxiety and Depression mean score differences across the study were the dependent variables. Overall fit of the model was $R^2 = 0.030$, p = 0.013 (Model A) and $R^2 = 0.040$, p = 0.003 (Model B). "Beta" is the standardized regression coefficient. Numbers in 'bold' indicate significant results. Backward regression method was used.

**Table 4. Student's responses regarding their learning process during the COVID-19 pandemic (n = 450).**

| Statements | Answer, n (%) | | | | |
|---|---|---|---|---|---|
| | Excellent | Very good | Good | Fair | Not good |
| How do you rank the online learning with regards to theory courses you received last semester compared to last year | 52 (11.6) | 71 (15.8) | 148 (32.9) | 64 (14.2) | 115 (25.6) |
| How do you rank the online learning with regards to practical courses you received last semester compared to last year | 35 (7.8) | 38 (8.4) | 104 (23.1) | 63 (14.0) | 210 (46.7) |
| How do you rank the online learning with regards to the assessment and exams you received last semester compared to last year | 49 (10.9) | 58 (12.9) | 153 (34.0) | 61 (13.6) | 129 (28.7) |
| How do you rank the learning with regards to your training last semester compared to last year | 35 (7.8) | 42 (9.3) | 129 (28.7) | 79 (17.6) | 165 (36.7) |
| How do you rank the online learning with regards to your relationship with your doctors last semester compared to last year's | 39 (8.7) | 51 (11.3) | 146 (32.4) | 85 (18.9) | 129 (28.7) |
| How do you rank the online learning with regards to its effect on your last semester Grade Point Average compared to last year's second semester | 66 (14.7) | 92 (20.4) | 148 (32.9) | 64 (14.2) | 80 (17.8) |

## Discussion

This study was the first to unveil the level of anxiety and depression of university healthcare students living in Jordan during the COVID-19 pandemic, in addition to unveiling the impact of the pandemic on their learning process. Students were found to be significantly affected mentally during the pandemic, with many showing borderline abnormal anxiety (22.4%) and depression (33.8%) symptoms. The learning process was also found to be negatively affected, as reported by 54.0% of the students. Certain variables were found to have a significant association with higher anxiety scores (worst anxiety level) including smoking, lower family income, and the use of medications by the students. The use of medications also showed a significant association with higher depression scores (worst depression level).

During the current global crisis and previous world pandemics, psychological and economic pressures increased [44], affecting families' and individuals' stability, which is a significant factor that can also affect students' anxiety levels. Emotional distress amongst students has been reported to be caused by the shortage in resources, forced social and physical distancing and self-quarantining, besides the social and financial losses [6, 11, 12]. The effect of social distancing and self-quarantining might leave young individuals, including students, feeling more vulnerable and lonelier leading to an increase of anxious and depressive symptoms [6]. Additionally, as distance learning became the route of education, which is a major change from the norm, it has been reported that the effect of the pandemic on students' studies, as well as their future employment, is related to their anxiety level [30, 31]. All of these factors provoked the increased levels of anxiety and depression among the students, affecting the physiological cortisol levels and the norm biological rhythms [21, 24, 25, 32, 33]. Several studies have been reporting increased levels of stress, anger, anxiety and depression among individuals worldwide due to the COVID-19 pandemic and public quarantine [11, 18, 19]. In Jordan, a state of emergency was declared by the government and a mandatory curfew was imposed on the 20th of March, and only started to ease off during May 2020. Hence, it was not surprising to identify through this study many medical students reporting anxiety and depression symptoms. Similar to the findings of this study, a recent study assessing the psychological impact of COVID-19 on university students reported that 34% of participants showed moderate to severe anxiety symptoms, while 28% of the students showed moderate to severe depression symptoms [34].

Several studies have reported the impact of COVID-19 not only on the anxiety and depression levels, but also on the sleep pattern among individuals [6, 31, 35]. Emotional distresses can lead to changes in sleep patterns and sleep difficulties, which have been reported among individuals and students who suffer from higher levels of stress, anxiety and depression [34–36]. This study confirms these results, as many of the participants (78.4%) reported changes in their sleep patterns. Female students were reported to be affected more severely with regards to their sleeping patterns and their psycho-emotional symptoms compared to males during the COVID-19 pandemic [34, 37]. This study showed no significant differences between females and males in this regard.

The need of understanding the impact of COVID-19 on the psychological aspect of students is crucial as the emerging need of psychological aid and mental healthcare is increasing. Through the multiple linear regression analysis, this study unveiled that the use of medications was the only variable positively associated with higher anxiety and depression scores. Other variables showed a significant association between smoking, lower family income and anxiety higher scores (more severe anxiety symptoms). A similar to a recent study which assessed the impact of the COVID-19 pandemic among college students in China found a similar result [25]. Sallam and his colleagues reported similar associations between smoking and lower-income to higher levels of anxiety among the residing-public in Jordan during the recent pandemic [38]. Drug-seeking and drug use may become more important among participants who are suffering higher levels of stress and anxiety, as much of the population during the pandemic are trying to manage and alleviate their negative stresses. Shortage and limited access to health care facilities and healthcare professionals are likely to exacerbate the levels of anxiety and stress especially among individuals with depressive symptoms leading to increased increments in the search and use of medications during the COVID-19 pandemic. A study conducted in Jordan looking into the level of psychological distress among university students (n = 381) during the pandemic showed that most of the respondents were regarded as having severe psychological distress (69.5%), with the most common coping strategy among the students involving spending more time on social media (70.6%) or using medications (12.9%) [37].

Emotional distress and higher anxiety levels are expected to be significantly associated with smokers compared to non-smokers, as they are reported to be individuals with higher comorbidities and vulnerability to get the virus with the possibility of a worsened prognosis [39, 40]. The higher risk on smokers along with the severe adverse outcomes such as needing mechanical ventilation or admission to ICU, can lead to stress and higher anxiety levels [39, 40].

When it comes to the tertiary education, the coronavirus pandemic is having a profound effect on all aspects of society, including the students' learning process. The COVID-19 pandemic has been significantly affecting the educational process and career progression of medical students [26]. Most governments postponed the activities and events which were planned to take place in educational institutions, in addition to initiating a complete closure of schools and universities, affecting 80% of the world's, and 100% of Jordan's, student population [20, 21]. This led to the implementation of distance learning as the main route of education in most countries, including Jordan [21–23]. Almost 54% of the students in this study ranked the learning process in Jordan during the pandemic as fair or not good. The majority agreed that the pandemic had a positive effect on their semester GPA, and their mean semester GPA total mark was slightly increased. Similarly, a significant positive impact of the COVID-19 was reported on students' performance improving the efficiency of the learning process and leading to better scores [41]. The reason behind this might be due to the increased chances of students getting help while answering their exams due to the proposed increased difficulty in monitoring the students during the exam time [42]. Several other challenges have been emerging with the online learning during this crisis era [21]. With regard to theory courses, many

reported the learning process as fair or not good (40%). Unfavourable study environment at home, poor internet connectivity and various problems related to anxiety and depression were reported by the students, leading to a significant negative impact on their learning process [43, 44]. Another study conducted in Jordan with the aim to assess the level of psychological distress among university students (n = 381) during the COVID-19 pandemic showed that distance learning was students' most serious concern (54.9%) and the majority (54.9%) reported that they had no motivation for distance learning [37]. A study conducted in the UK during the pandemic looking into its impact on the education sector after many universities halted campus-based teaching and examinations revealed negative impact on final year medical students' (n = 440) confidence and preparedness as they went into their first year of foundation training [45].

The HADS assessment was used in this study revealing students' anxiety and depression levels during this pandemic. The validity, reliability and availability of the HADS questionnaire as published in many countries make it the most suitable tool for bi-dimensional assessment of anxiety and depression symptoms amongst the study participants [28]. In addition, due to the small number of questions it comprises (14 items, seven relate to anxiety symptoms and seven to depression symptoms), the questionnaire is feasible when completed online. Also, as each item on the questionnaire is scored from 0–3, this tool provides a score of anxiety and a score of depression (can vary from 0–21). Through reviewing a large number of literature, the cut-off point of 8/21 for anxiety or depression has been identified [28].

This study comes with some limitations. With the methodology involving an online questionnaire, it meant that participants were not met face-to-face, and the response rate could not be assessed. However, during the pandemic and the public curfew imposed in Jordan in response to the emergency situation, this was the only possible way to run this study. Representativeness with regards to participants' profile is another limitation considering that most participants were from universities located in the capital Amman or Irbid, the second-largest Jordanian metropolitan after Amman. Also, the respondents self-selected themselves in choosing to answer the survey, which could be considered a selection bias. The survey consisted of parts which were not validated, including the assessment of participants' demographics and effects on their learning process. Nevertheless, these parts were put together in real-time, based on current literature and reviewed by an expert team of clinical pharmacists and academics before the study.

In conclusion, this study is the first to highpoint that healthcare students from numerous disciplines and universities in Jordan report to have symptoms of anxiety and depression during the COVID-19 pandemic. Outcomes show that the learning process during the pandemic has also been affected, with the majority of students reporting dissatisfaction with the online learning process that replaced the face-to-face educational process followed prior to the pandemic in the universities. Results of this study have important international applicability, as the universities worldwide shared comparable changes during the recent pandemic, specifically with regards to the taken measures with regards to turning to distant online healthcare education and training [46]. Findings of this study call onto the policymakers and healthcare educators at the universities to implement different interventions and conduct workshops targeting the mental health of healthcare students, and to identify and implement resolutions that aim to optimize the learning process during the pandemic and maintain its efficiency.

## Supporting information

**S1 Appendix. Prevalence of anxiety and depression during COVID-19 pandemic among healthcare students in Jordan survey.**
(DOCX)

## Author Contributions

**Formal analysis:** Iman A. Basheti.

**Investigation:** Qassim N. Mhaidat, Hala N. Mhaidat.

**Methodology:** Iman A. Basheti, Qassim N. Mhaidat, Hala N. Mhaidat.

**Supervision:** Iman A. Basheti.

**Writing – original draft:** Qassim N. Mhaidat, Hala N. Mhaidat.

**Writing – review & editing:** Iman A. Basheti.

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
