## [Decision Letter · Decision Letter 0]

26 Jan 2021

PONE-D-20-26311

Prevalence of anxiety and depression following COVID-19 pandemic among healthcare students in Jordan and its effect on their learning process: a national survey

PLOS ONE

Dear Dr. Basheti,

Thank you for submitting your manuscript to PLOS ONE. After careful consideration, we feel that it has merit but does not fully meet PLOS ONE’s publication criteria as it currently stands. Therefore, we invite you to submit a revised version of the manuscript that addresses the points raised during the review process. The  expert reviewer raised concerns regarding the questionnaire, the discussion and the response rate among others that need to be addressed.

We look forward to receiving your revised manuscript.

Kind regards,

Mohammed Saqr, Ph.D

Academic Editor

PLOS ONE

Journal Requirements:

2. Please ensure that your conclusions are supported by data. For example, remove any suggestions that the anxiety levels recorded are a consequence of the pandemic since pre-pandemic anxiety levels were not measured.

Reviewers' comments:

Reviewer's Responses to Questions

**Comments to the Author**

1. Is the manuscript technically sound, and do the data support the conclusions?

Reviewer #1: Yes

2. Has the statistical analysis been performed appropriately and rigorously? 

Reviewer #1: Yes

3. Have the authors made all data underlying the findings in their manuscript fully available?

Reviewer #1: Yes

4. Is the manuscript presented in an intelligible fashion and written in standard English?

Reviewer #1: Yes

5. Review Comments to the Author

Reviewer #1: The authors in this study explored the impact of COVID-19 on healthcare students in several universities in Jordan using an online survey. The pandemic had a negative impact on students and their learning process.

However, I have the following comments and concerns:

1- The response rate was not calculated in this study, I believe this should be stated. Also, what is the estimated number of healthcare students in this year in Jordan. The sample might not be truly representative.

2- Respondents self-selected themselves in the survey which could be considered a selection bias

3- Was the survey tested in a pilot study ?

4- The sentences in the Introduction from line (97-104) needs to be rephrased (some errors: China, severe ..needs to be corrected, it should be clear that SARS-CoV 2 is the virus name, and COVID-19 is the disease that was declared a pandemic)

5- Authors can add a sentence regarding the current status of COVID-19 infection in Jordan, and the timing of quarantine.

6- The graph (Figure 1) quality is poor and needs to be enhanced

7- I think the discusson can be enriched with the following references:

Choi, B., Jegatheeswaran, L., Minocha, A. et al. The impact of the COVID-19 pandemic on final year medical students in the United Kingdom: a national survey. BMC Med Educ 20, 206 (2020). https://doi.org/10.1186/s12909-020-02117-1

Is it Just About Physical Health? An Internet-Based Cross-Sectional Study Exploring the Psychological Impacts of COVID-19 Pandemic on University Students in Jordan Using Kessler Psychological Distress Scale Ala'a B. Al-Tammemi, Amal Akour, Laith Alfalah medRxiv 2020.05.14.20102343; doi: https://doi.org/10.1101/2020.05.14.20102343

6. PLOS authors have the option to publish the peer review history of their article (what does this mean?). If published, this will include your full peer review and any attached files.

Reviewer #1: **Yes: **Ismail Ibrahim Ismail

---

## [Author Response · Author response to Decision Letter 0]

2 Mar 2021

Revision for the manuscript number ‘PONE-D-20-26311’ titled ‘Prevalence of anxiety and depression following COVID-19 pandemic among healthcare students in Jordan and its effect on their learning process: a national survey’, to be published in PLOS ONE.

Dear Dr. Mohammed Saqr, Ph.D.

Academic Editor, PLOS ONE.

We would like to thank you and the reviewer for your evaluation and suggestions about the manuscript. Please find attached our point-by-point responses to the reviewer’s comments. Revised/additional text within the manuscript has been marked with tracked changes, and a clean copy has been uploaded. 

Editor’s comments:

C1. Please ensure that your manuscript meets PLOS ONE's style requirements, including those for file naming. The PLOS ONE style templates can be found at 

 R1. Done.

C2. Please ensure that your conclusions are supported by data. For example, remove any suggestions that the anxiety levels recorded are a consequence of the pandemic since pre-pandemic anxiety levels were not measured.

R2. Done. The conclusion now reads as follows:

Abstract:

‘Conclusions: Anxiety and depression levels amongst university healthcare students in Jordan were found to be high when assessed during the COVID-19 pandemic. In addition, the learning process during the pandemic was not accepted by more than half of the students. Implementing psychological interventions for healthcare students during pandemics is strongly recommended in order to optimize students’ mental health and their learning process alike.’ 

Manuscript:

‘In conclusion, this study is the first to highpoint that healthcare students from numerous disciplines and universities in Jordan report to have symptoms of anxiety and depression during the COVID-19 pandemic. Outcomes show that the learning process during the pandemic has also been affected, with the majority of students reporting dissatisfaction with the online learning process that replaced the face-to-face educational process followed prior to the pandemic in the universities. Results of this study have important international applicability, as the universities worldwide shared comparable changes during the recent pandemic, specifically with regards to the taken measures with regards to turning to distant online healthcare education and training [46]. Findings of this study call onto the policymakers and healthcare educators at the universities to implement different interventions and conduct workshops targeting the mental health of healthcare students, and to identify and implement resolutions that aim to optimize the learning process during the pandemic and maintain its efficiency.’

Reviewers' comments:

The authors in this study explored the impact of COVID-19 on healthcare students in several universities in Jordan using an online survey. The pandemic had a negative impact on students and their learning process. However, I have the following comments and concerns:

C1. The response rate was not calculated in this study, I believe this should be stated.

R1. The following was added to the limitation section:

‘This study comes with some limitations. With the methodology involving an online questionnaire, it meant that participants were not met face-to-face, and the response rate could not be assessed.’

C2. Also, what is the estimated number of healthcare students in this year in Jordan. The sample might not be truly representative.

R2. The following sample size calculation section was added to the manuscript:

‘Sample size

According to the Ministry of Higher Education and Scientific Research, the number of the healthcare students in Jordan in 2020 was 16, 214. Based on this number, the sample size was calculated using a margin of error of 5%, confidence level of 95%, and response distribution of 50%, to be a minimum of 375 students.’ 

C3. Respondents self-selected themselves in the survey which could be considered a selection bias.

R3. The following comment was added to the limitation section:

‘Also, the respondents self-selected themselves in choosing to answer the survey, which could be considered a selection bias.’

C4. Was the survey tested in a pilot study?

R4. Yes. The following sentence was added to the methodology section.

‘Following development by the research team, the online survey was tested during a pilot phase of this study, which was conducted over three days before the actual study data collection period was initiated. The online survey was completed by students (n= 5) and academics (n= 5) who gave their feedback and recommendations regarding the survey. The research team studies these recommendations and feedback provided, and a final version of the survey was prepared before the study started. Results of the pilot phase of the study were not included in the analysis of this study.’

C5. The sentences in the Introduction from line (97-104) needs to be rephrased (some errors: China, severe ..needs to be corrected, it should be clear that SARS-CoV 2 is the virus name, and COVID-19 is the disease that was declared a pandemic).

R5. The sentence was corrected as follows:

‘In March 2020, the World Health Organization (WHO) announced the corona virus disease, termed COVID-19 disease, as an international pandemic, which was caused by the infectious virus ‘severe acute respiratory syndrome (SARS) coronavirus 2’’

C6. Authors can add a sentence regarding the current status of COVID-19 infection in Jordan, and the timing of quarantine.

R6. The following sentence was added to the introduction:

‘On July 29, 2020, the Jordan Ministry Of Health reported that the COVID-19 outbreak has resulted in 1,187 confirmed cases in Jordan, and 11 cases of confirmed deaths [8].’

C7. The graph (Figure 1) quality is poor and needs to be enhanced.

R7. Done. 

C8. I think the discussion can be enriched with the following references:

- Choi, B., Jegatheeswaran, L., Minocha, A. et al. The impact of the COVID-19 pandemic on final year medical students in the United Kingdom: a national survey. BMC Med Educ 20, 206 (2020). https://doi.org/10.1186/s12909-020-02117-1

- Is it Just About Physical Health? An Internet-Based Cross-Sectional Study Exploring the Psychological Impacts of COVID-19 Pandemic on University Students in Jordan Using Kessler Psychological Distress Scale Ala'a B. Al-Tammemi, Amal Akour, Laith Alfalah medRxiv 2020.05.14.20102343; doi: https://doi.org/10.1101/2020.05.14.20102343.

R8. These references were used to update the discussion of the manuscript as shown.

---

## [Decision Letter · Decision Letter 1]

24 Mar 2021

Prevalence of anxiety and depression during COVID-19 pandemic among healthcare students in Jordan and its effect on their learning process: a national survey

PONE-D-20-26311R1

Dear Dr. Basheti,

We’re pleased to inform you that your manuscript has been judged scientifically suitable for publication and will be formally accepted for publication once it meets all outstanding technical requirements.

Kind regards,

Mohammed Saqr, Ph.D

Academic Editor

PLOS ONE

Additional Editor Comments (optional):

Reviewers' comments:

Reviewer's Responses to Questions

**Comments to the Author**

1. If the authors have adequately addressed your comments raised in a previous round of review and you feel that this manuscript is now acceptable for publication, you may indicate that here to bypass the “Comments to the Author” section, enter your conflict of interest statement in the “Confidential to Editor” section, and submit your "Accept" recommendation.

Reviewer #1: All comments have been addressed

2. Is the manuscript technically sound, and do the data support the conclusions?

Reviewer #1: Yes

3. Has the statistical analysis been performed appropriately and rigorously? 

Reviewer #1: I Don't Know

4. Have the authors made all data underlying the findings in their manuscript fully available?

Reviewer #1: Yes

5. Is the manuscript presented in an intelligible fashion and written in standard English?

Reviewer #1: Yes

6. Review Comments to the Author

Reviewer #1: All comments have been addressed properly. The authors added most of the comments to the Limitations section in the study. They also updated the manuscript with the suggested references. I have no further recommendations.

7. PLOS authors have the option to publish the peer review history of their article (what does this mean?). If published, this will include your full peer review and any attached files.

Reviewer #1: **Yes: **Ismail Ibrahim Ismail

---

## [Editor Report · Acceptance letter]

26 Mar 2021

PONE-D-20-26311R1 

Prevalence of anxiety and depression during COVID-19 pandemic among healthcare students in Jordan and its effect on their learning process: a national survey 

Dear Dr. Basheti:

I'm pleased to inform you that your manuscript has been deemed suitable for publication in PLOS ONE. Congratulations! Your manuscript is now with our production department. 

Kind regards, 

on behalf of

Dr. Mohammed Saqr 

Academic Editor

PLOS ONE